# SaraCoder: Orchestrating Semantic and Structural Cues for Resource-Optimized Repository-Level Code Completion

## Abstract

Despite Retrieval-Augmented Generation improving code completion, traditional retrieval methods struggle with information redundancy and a lack of diversity within limited context windows. To solve this, we propose a resource-optimized retrieval augmentation method, SaraCoder. It maximizes information diversity and representativeness in a limited context window, significantly boosting the accuracy and reliability of repository-level code completion. Its core Hierarchical Feature Optimization module systematically refines candidates by distilling deep semantic relationships, pruning exact duplicates, assessing structural similarity with a novel graph-based metric that weighs edits by their topological importance, and reranking results to maximize both relevance and diversity. Furthermore, an External-Aware Identifier Disambiguator module accurately resolves cross-file symbol ambiguity via dependency analysis. Extensive experiments on the challenging CrossCodeEval and RepoEval-Updated benchmarks demonstrate that SaraCoder outperforms existing baselines across multiple programming languages and models. Our work proves that systematically refining retrieval results across multiple dimensions provides a new paradigm for building more accurate and resource-optimized repository-level code completion systems.

## 1 Introduction

Code Large Language Models (Code LLMs) Izadi et al. (2022); Li et al. (2022b); Allal et al. (2023), built on the Transformer architecture and trained on massive code corpora. These models compress vast programming knowledge into hundreds of millions of parameters and have been successfully applied in numerous real-world development scenarios, and have significantly advanced the intelligence of modern software development. However, these models often only process local information within their context window. As codebases grow and development tasks become increasingly complex, this limitation becomes more pronounced. Tasks like understanding functionality and fixing bugs often require integrating a wide range of context, such as related API definitions, dependent modules, and type constraints. Since the models cannot perceive information outside their window, the suggestions they generate tend to be localized, often insufficient, and inaccurate.

Retrieval-Augmented Generation (RAG) Tang et al. (2023); Zan et al. (2022); Zhang et al. (2023) overcomes the limitations of traditional models by introducing an external knowledge retrieval mechanism. In this framework, an efficient retriever can dynamically fetch relevant code snippets, API documentation, or type definitions from an external codebase in real time, thereby expanding the model's contextual awareness. The generative model then integrates these retrieval results with the current context to produce syntactically correct, semantically consistent, and standard-compliant code suggestions. Retrieval-Augmented Generation is gradually becoming a foundational technology for achieving accurate and trustworthy repository-level intelligent code completion.

However, current Retrieval-Augmented Generation (RAG) methods for code completion often have a single-aspect information problem. Methods like GraphCoder Liu et al. (2024b) focus on retrieving similar code snippets, while others like DraCo Cheng et al. (2024) prioritize cross-file context. This narrow focus overlooks the need to combine different types of information, even though developers in real-world settings require both cross-file context for project-wide consistency and similar

code snippets for structural reference and efficiency. Limited resources, such as a finite context window, make the effective fusion of these different information types a critical challenge. To make matters worse, existing similarity-based retrieval methods have three major practical flaws. **(a)** *Misleading Superficial Similarity*: They often retrieve irrelevant code snippets that share surface-level similarities, which can mislead the model and result in incorrect or ineffective suggestions. **(b)** *Redundant Retrieval Homogenization*: Relying solely on similarity ranking frequently yields redundant or duplicate snippets. This wastes valuable context window space and provides a homogeneous, narrow viewpoint, hindering the model's ability to generate innovative or superior solutions. **(c)** *External Symbol Ambiguity*: These methods fail to capture essential dependencies like unreferenced classes or architectural rules. This ambiguity can trigger a cascade of errors, including failed type inference and mismatched signatures, compromising the correctness and reliability of the generated code. Ultimately, these flaws not only degrade the quality of retrieved information but also exacerbate the challenge of a finite context window by filling the limited space with irrelevant, repetitive, or ambiguous data, thereby compromising the generation of correct and innovative code.

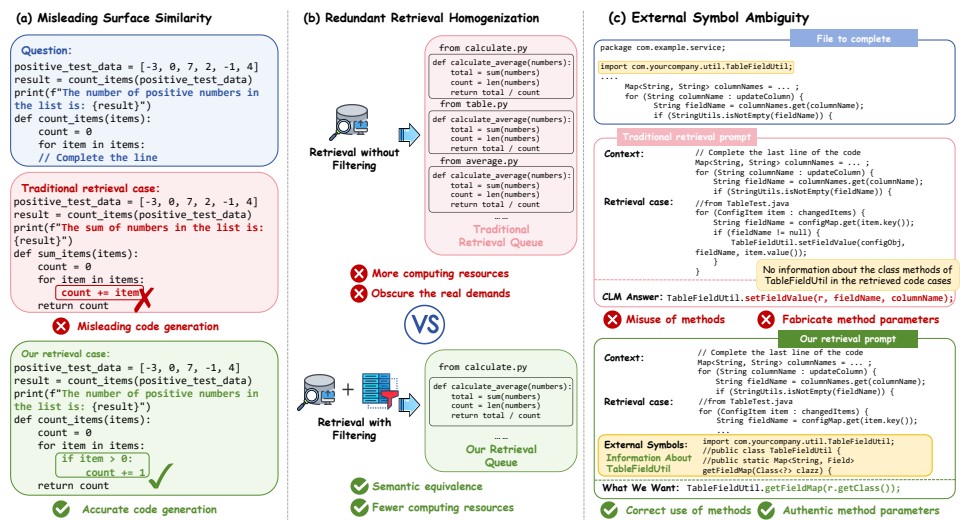

Figure 1: The pitfalls of pure similarity retrieval and the highlights of SARACODER. Pink boxes illustrate traditional retrieval results based purely on surface similarity, while green boxes demonstrate results from our method SARACODER.

To address these challenges, we introduce SARACODER, a **S**emantic-**A**ware Code **R**etrieval framework with **R**edundancy Reduction and **A**mbiguity Suppression for Repository-Level Code Completion, that leverages hierarchical resource-optimization to enhance repository-level code completion under resource-constrained conditions. By deeply mining relationships in code snippets, removing redundancy and optimized reranking, we provide LLMs with richer, higher-quality reference cases while minimizing context window consumption. To address misleading superficial similarity, we utilize semantic alignment distillation to capture deep semantic relationships and a graph-based structural similarity metric, which weighs editing operations by topological importance to assess the structural proximity of candidates to the target context. To combat redundant retrieval homogenization, we integrate MD5-based deduplication pruning and diversity-aware reranking, ensuring relevance while maximizing diversity. Additionally, to resolve external symbol ambiguity in repository-level code completion, we introduce an external-aware identifier disambiguator that analyzes project-level dependencies for LLMs. Our key contributions are:

- We propose SARACODER, a hierarchical and resource-optimized retrieval-augmented code completion framework. SARACODER has redefined its code completion goal from retrieving a large volume of relevant code to providing the most valuable information within a limited context.

- As an intelligent information filter, SARACODER upgrades code completion by moving from surface-level code matching to intelligent decision-making based on deep semantics and project structure. This provides LLMs with high-quality, precise information, leading to more efficient repository-level code completion within limited contexts.

- SARACODER's design allows it to maintain suggestion quality even with a nearly full context window. This ensures that when complementing other methods that provide different information, it minimizes negative interference and maximizes the preservation of their content. This synergy enables it to complement orthogonally other methods that provide cross-file context, providing a cooperative improvements when used in combination.

## 2 RELATED WORK

Current Retrieval-Augmented Generation methods for repo-level code completion mainly rely on code similarity or cross-file dependencies.

### 2.1 SIMILAR CODE SNIPPET RETRIEVAL FOR RAG

This approach enhances the quality of code LLM generation by retrieving semantically similar code snippets and integrating them into prompts, mimicking the reference behaviors of programmers. CodeSearchNet Husain et al. (2020) pioneers large-scale code corpora construction, providing retrieval-based completion references; CodeRetriever Li et al. (2022a) integrates pretrained models like CodeBERT Feng et al. (2020) to enhance complex scenario handling; ReACC Lu et al. (2022) combines vector and string-based retrieval to significantly optimize long-code processing; Graph-Coder Liu et al. (2024b) improves code completion by using program dependencies for structured representations, allowing coarse-to-fine retrieval for Python and Java. However, like many other methods, its dependency analysis does not fully grasp deep semantic relationships in code. Additionally, most approaches rely too much on surface-level textual similarity. This often results in redundant retrieved content, wasting resources.

### 2.2 CROSS-FILE DEPENDENCY RETRIEVAL AUGMENTATION

This method approaches code completion in complex repositories by leveraging cross-file code context (e.g., dependencies, dataflow, and subsequent similar code). Inspired by Ding et al.'s Ding et al. (2023) observation that subsequent content of high-similarity snippets effectively informs completion, it injects these snippets into prompts. COCOMIC Ding et al. (2024) dynamically fuses the context of this file with the cross-file entities retrieved by CCFinder (compressed into [SUM] vectors) through a joint attention mechanism to achieve location-aware code completion. DraCo Cheng et al. (2024) extends this paradigm through dataflow-guided retrieval, parsing private repositories into code entities and constructing a repository-specific context graph reflecting dependencies. DraCo retrieves precise contextual knowledge from this graph to generate well-structured prompts, overcoming cross-file information barriers and repository-specific accuracy gaps. Current limitations include Python-exclusive implementation with type-sensitive dependencies lacking multilingual support.

### 2.3 REPOSITORY-LEVEL CODE COMPLETION EVALUATION

Traditional code completion benchmarks like Chen et al. (2021); Austin et al. (2021) focus on isolated snippets, but modern software development's complexity demands a better evaluation. To address this, specialized benchmarks such as RepoEval Zhang et al. (2023), CrossCodeEval Ding et al. (2023), RepoBench Liu et al. (2024a), and ReccEval Cheng et al. (2024) have emerged. They provide standardized, rigorous tests across various languages and project scales. These benchmarks divide repository-level code completion into two scenarios. **(1) In-File Completion:** A high-frequency task that uses only the current file's context (e.g., RepoEval). **(2) Cross-File Completion:** A more complex task that requires understanding and completing code with dependencies on symbols from other files (e.g., CrossCodeEval and ReccEval). This shift highlights the move from simple, isolated tests to comprehensive evaluations that reflect real-world coding environments.

## 3 METHOD

As shown in Figure 2, SARACODER is a **hierarchical feature-optimized retrieval-enhanced code completion framework**. Formally, given a code context $C_{context} = \{x_1, x_2, \cdots, x_n\}$ and its containing file path $F$, the task aims to predict the next statement $\tilde{y}$.

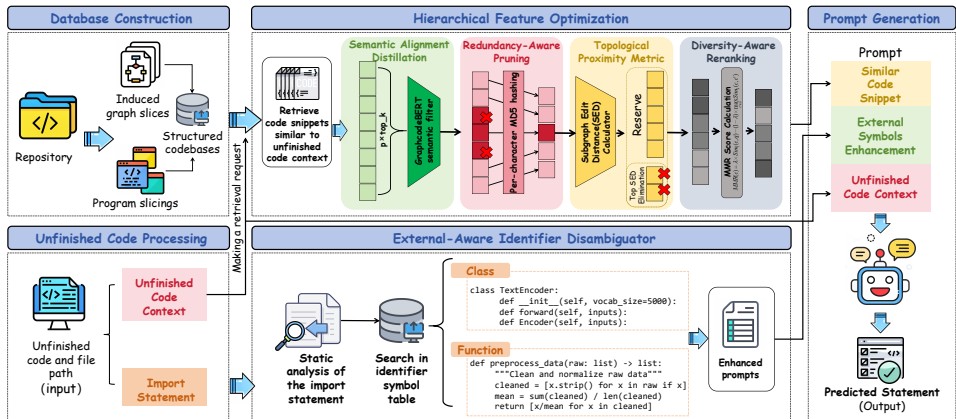

Figure 2: An illustration of SARACODER framework. **(1) Database Construction.** This phase constructs a key-value codebase. This involves using a slicing algorithm to create induced graph slices, which are then precisely mapped to source code snippets. **(2) Code Retrieval.** This phase takes code context as input and retrieves similar code, then refines suggestions via Hierarchical Feature Optimization. Concurrently, an External-Aware Identifier Disambiguator clarifies external symbols via dependency analysis, delivering highly accurate candidates. **(3) Code Generation.** This phase generates prompts by integrating outputs from code retrieval with the code completion context. These prompts are then fed into an LLM to predict completion statements.

## 3.1 DATABASE AND INITIAL CANDIDATE CONSTRUCTION

To better represent code logic, we introduce a multi-level code context graph model that integrates control flow, data dependency, and control dependency Liu et al. (2024b). This structured representation offers enhanced generalization capabilities compared to serialization methods, enabling more effective capture of task-relevant context and facilitating easier adaptation to other languages. We utilize program slicing to generate precisely mapped, task-relevant subgraphs from source code on-demand, constructing structured codebases tailored to support specific analysis tasks. When a code completion task request occurs, we extract the Unfinished Code Context $C_{context}$ and the Import Statements $I$ from the code file . $C_{context}$ is then used to retrieve an initial candidate set of $top\_k \times p$ [1] code snippets $C$ via text similarity from structured codebases.

## 3.2 HIERARCHICAL FEATURE OPTIMIZATION (HF_OP)

### 3.2.1 SEMANTIC ALIGNMENT DISTILLATION

Semantic alignment distillation addresses Superficial Similarity Misguidance by leveraging the GraphcodeBERT Guo et al. (2021), a pretrained model specialized in code understanding, to capture deep semantic relationships between code snippets. First, the query code $Q$ and candidate set $C$ are tokenized into subword sequences and uniformly padded or truncated to a fixed length $L = 512$. Subsequently, during the feature encoding phase, a 768-dimensional semantic vector $v_s$ is extracted for each code unit $s \in Q \cup C$, with vector space standardized through L2 normalization. When code repositories lack sufficient repetitive or relevant code, standard filtering methods are too strict, often leading to **zero-candidate scenarios**. This scarcity of reference material then hurts the accuracy of large language models. To fix this "one-size-fits-all" problem, we introduce a new dynamic quantile threshold mechanism. During the dynamic filtering phase, the cosine similarity set $S = \{\cos(v_Q, v_c) \mid c \in C\}$ is computed between the query vector $v_Q$ and all candidate vectors $v_c$. An adaptive threshold $\tau = \text{quantile}(S, 0.75)$ is set at the 75th percentile, outputting filtered results $C_{SAD} = \{c \mid \cos(v_Q, v_c) \geq \tau\}$. To reduce redundant computation overhead and improve efficiency, a caching mechanism stores encoding results for high-frequency code.

---

[1]To ensure a high-quality final candidate set of $top\_k$ results, we expand the initial candidate pool to $top\_k \times p$, allowing more candidates to participate in the Hierarchical Feature Optimization.

### 3.2.2 REDUNDANCY-AWARE PRUNING

This module implements lightweight hash-based deduplication via exact text matching. Using the MD5 algorithm Rivest (1992), it generates 128-bit hash fingerprints (*single computation* $\approx 0.02$ms, *memory footprint* 32 bytes/hash) to eliminate verbatim duplicates from candidate set $C_{SAD}$ with minimal computational cost, significantly reducing downstream overhead. The module maintains a global hash set $H_{\text{seen}}$ to dynamically track processed sample fingerprints: for each candidate $c \in C_{SAD}$, if its MD5 hash $h_c \notin H_{\text{seen}}$, $c$ is added to deduplicated result set $C_{RAP}$ and $H_{\text{seen}}$ is updated. This achieves real-time processing with $O(N)$ time complexity. After semantic alignment distillation processing, the number of code snippets requiring MD5 hashing is limited and their structure is fixed by syntactic and semantic constraints. The MD5 collision resistance (theoretical probability $\approx 1.47 \times 10^{-18}$) is sufficient for strict sensitivity. Additionally, MD5's superior speed and lower memory footprint provide optimal cost-performance.

### 3.2.3 TOPOLOGICAL PROXIMITY METRIC

At this layer, the decaying subgraph edit distance (D-SED) is introduced to measure the graph similarity between the query graph $G_q$ and the candidate graph $G_c$ (Ranjan et al.; Zeng et al.). A higher D-SED value indicates less similarity. We calculate D-SED for code snippets to quantify their structural similarity and retain those with the closest match.

$$D - \text{SED}\,(G_q, G_c) = \sum_{op=\text{O}} \gamma^{l(op)} \cdot c(op) \tag{1}$$

Editing operations $O$ are the set of operations to transform $G_c$ to $G_q$, include adding, deleting, and modifying nodes and edges. Each operation $op \in O$ has a cost $c(op)$ and a hop count $l(op)$ from its "core node". For simplicity, we choose the node with the largest ID as core node. Operations closer to the core exert greater structural influence. $\gamma \in (0, 1)$ is an attenuation factor that reduces the cost weight for operations farther from the core node. After computing D-SED scores for each candidate $c \in C_{RAP}$, we compute a composite score $s$ as a weighted sum of text similarity (calculated during initial candidate generation) and structural similarity (D-SED scores). Subsequently, we generate $Q_{TPM} = [(\text{c}, \text{s}), \ldots]$, ordered in descending score $s$.

### 3.2.4 DIVERSITY-AWARE RERANKING

This module implements a variability-aware ranking model based on the Maximal Marginal Relevance (MMR) Carbonell & Goldstein (2017) algorithm to maximize result diversity while preserving relevance. It addresses homogeneity in traditional rankings through adversarial similarity calculation and dynamic weight adjustment. $S$ contains items $(c, s) \in Q_{TPM}$ that have not been selected into $C_{final}$ yet. $\text{Sim}_1$ represents the relevance ($s_i = \pi_2 \circ \iota_{c_i}(S)$) of item $c_i$ to query $q$. $\text{Sim}_2$ denotes the maximum cosine similarity between $c_i$ and any item $c_j$ in the selected set $C_{final}$. $\lambda$ is a trade-off parameter that balances the emphasis between relevance ($\text{Sim}_1$) and diversity ($\text{Sim}_2$).

$$\text{MMR} = \arg\max_{c_i \in S} \left[ \lambda \cdot \text{Sim}_1(c_i, q) - (1 - \lambda) \cdot \max_{c_j \in C_{\text{final}}} \text{Sim}_2(c_i, c_j) \right] \tag{2}$$

### 3.3 EXTERNAL-AWARE IDENTIFIER DISAMBIGUATOR (EAID)

This module enhances knowledge through external identifier augmentation. Firstly, file-level entity modeling parses code per file $F$, extracting method entities $E_{\text{method}}$ (functions/class methods with identifier, alias, line range $[l_{\text{start}}, l_{\text{end}}]$, parameter signature, scope) and class entities $E_{\text{class}}$ (class definitions with identifier, alias, line range $[l_{\text{start}}, l_{\text{end}}]$, member mappings) that built in the $F$. After that, it generates a structured identifier symbol table $ST_{\text{lib}} = \{\text{identifier} \mapsto \text{syntax features}\}$, where *identifier* corresponds to either: (1) the unique identifier of a method entity $\forall e \in E_{\text{method}}$, or (2) the unique identifier of a class entity $\forall c \in E_{\text{class}}$, with the mapped *syntax features* containing all associated attributes for that entity. Subsequently, the dependency resolution mechanism processes all import statements ($I$) within the unfinshed file. For Intra-project Cross-module Reference, this phase retrieves complete entities ($E_{lib}$) from the pre-built entity library ($ST_{\text{lib}}$) by determining their corresponding file paths ($p$). These paths are constructed through decomposition of module components derived from either dotted names (e.g., `my.module.MyClass`) or relative imports (e.g., `from`

.sub_module import MyClass), which are subsequently joined using directory separators (/) and appended with the .py file extension. For standard and third-party libraries, the system constructs a lightweight reference table $T_{ext} = \{$canonical name $\mapsto$ alias$\}$ to efficiently manage external dependencies without full entity resolution. The enhanced prompts $P_E = I \oplus E_{lib} \oplus T_{ext}$.

## 3.4 PROMPT GENERATION

Following code retrieval and external link resolution, SARACODER employs an external LLM to generate subsequent statements. The final prompt $P_{final}$ is constructed by concatenating three components: the **external symbols enhancement** $P_E$ where entities are ordered by file import sequence reflecting call probability decay—with function entities populated with complete function bodies and class entities containing variable tables and method definitions; the **similar code snippets** $C_{final}$ containing code snippets strictly sorted in ascending order of similarity and annotated with source paths; the **unfinished code context** $C_{context}$. This architecture follows $P_{final} = C_{final} \oplus P_E \oplus C_{context}$. (Figure 3).

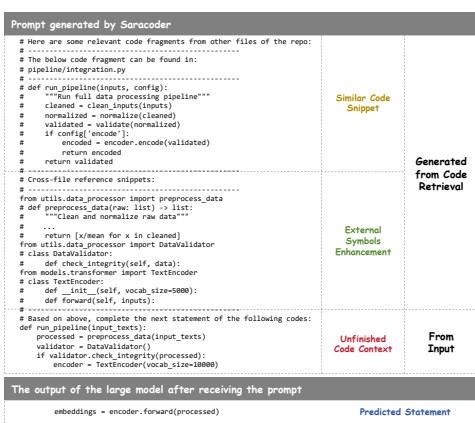

Figure 3: Prompt template.

## 4 EXPERIMENTS

### 4.1 EXPERIMENTAL SETTINGS

#### 4.1.1 DATASETS

We primarily utilize two datasets here: CrossCodeEval and RepoEval-Updated. **(1) CrossCodeEval Ding et al. (2023):** This benchmark evaluates code completion in complex Cross-File scenarios like type inference and dependency analysis. It is ideal for assessing performance that requires a deep understanding of code across multiple files. **(2) RepoEval-Updated Liu et al. (2024b):** Expanded from RepoEval Zhang et al. (2023), this new version, includes repositories of varying scales, offering

Table 1: CrossCodeEval vs. RepoEval-Updated comparison.

|  | CrossCodeEval | | RepoEval-Updated | |
|---|---|---|---|---|
|  | **Python** | **Java** | **Python** | **Java** |
| Total Repositories | 471 | 239 | 10 | 8 |
| Total Files | 14348 | 5868 | 3258 | 8260 |
| Total Task cases | 2665 | 2139 | 2000 | 1600 |
| Applicable Scenarios | Cross-File completion | | In-File completion | |

a better way to evaluate In-File completion performance. We use CrossCodeEval to test models on code completion tasks required complex cross-file dependencies. RepoEval-Updated assesses basic syntax, common API usage, and local context understanding. Table 1 shows details.

#### 4.1.2 EVALUATION INDICATORS SETTING

In this study, the following several evaluation indicators are used to assess the effect of code completion Lu et al. (2021); Ding et al. (2023).

- **Code Exact Match (EM):** Proportion of generated code exactly matching the ground truth. EM is given only for a perfect semantic and syntactic match.
- **Identifier Exact Match (ID_EM):** The percentage of identifiers (variables, functions, etc.) perfectly matching the ground code. A high ID_EM score indicates the model's strong contextual understanding, enabling it to accurately predict and generate contextually appropriate identifiers.
- **Identifier F1 Score (ID_F1):** A more nuanced evaluation of identifier matching by combining precision and recall. It offers a more comprehensive assessment of identifier completion quality, particularly beneficial in scenarios where models might generate partial but correct identifier sets.
- **Edit Similarity (ES):** Similarity metric between generated and ground-truth code based on edit distance. It tolerates slight variations, requiring the completed code to be highly similar in structure, syntax, and token order to the target.

### 4.1.3 BASELINE SETTING

We employ the following five methods as controls to evaluate the effectiveness of retrieval-augmented generation (RAG) in code completion: No RAG (Zero-shot-only baseline), Shifted RAG (Target-context dynamic retrieval), Vanilla RAG (Exemplar-similarity fixed retrieval), Repocoder (Iterative-fragment integration Zhang et al. (2023)), Graphcoder (Structure-modeling CCG utilization Liu et al. (2024b)).

### 4.1.4 MODEL SELECTION

In this experiment, we select `Codegen2-7b`, `Codegen25-7b` Nijkamp et al. (2023b;a), `CodeLlama-7b-Instruct` Rozière et al. (2024) and `deepseek-coder-6.7b-instruct` Guo et al. (2024) for code completion task inference.

## 4.2 MAIN RESULTS

To evaluate the performance of SARACODER on repository-level code completion, we have formulated the following four research questions (RQs):

**RQ1 Effectiveness in Cross-File Scenarios:** How does SARACODER perform when cross-context understanding is required, compared to other methods?

**RQ2 Cost Analysis in Cross-File Scenarios:** How does SARACODER's resource consumption compare to GraphCoder in Cross-File scenarios?

**RQ3 Synergistic Gain Property:** How does SARACODER perform when integrated orthogonally with other methods that provide cross-file context?

**RQ4 Advantage in In-File Scenarios:** How does SARACODER perform on tasks without cross-context requirements and what are its advantages?

### 4.2.1 FOR RQ1: DOMINANT CROSS-FILE CODE ACCURACY.

Table 2 illustrates that SARACODER surpasses the top-performing Repocoder on the CrossCodeEval dataset, achieving an average improvement of 1.50 in EM, 0.77 in ES, 1.11 in ID_EM, and 0.61 in ID_F1. This indicates SARACODER provides more effective information and generates code with higher semantic accuracy, better capturing intended functionality. The enhanced ID_EM further shows SARACODER's superior ability to interpret context and select appropriate identifiers. These advancements effectively mitigate misleading superficial similarity and external symbol ambiguity, leading to more reliable and contextually relevant code. For Java code completion, SARACODER shows better EM and ID_EM, with slightly lower ES and F1 scores. This discrepancy is attributed to the inherent characteristics of Java's static typing system and complex code structure. These features lead to the generation of code that is logically correct but contains numerous textual variations and boilerplate redundancies. Consequently, small structural deviations (e.g., misplaced brackets) are more readily penalized by metrics such as ES and F1.

### 4.2.2 FOR RQ2: COST-OPTIMIZED ACCURACY ADVANTAGE IN CROSS-FILE.

We experiment with code completion efficiency using codegen25-7b and Graphcoder. Our goal is to see how retrieving more similar cases ($top\_k$) impacts accuracy. Since using fewer $top\_k$ cases saves input tokens[2], this study shows the balance between resources and accuracy. Our experiments demonstrate significant performance saturation for both retrieval methods when $top\_k$ reaches 3-4, with no observable fluctuations upon increasing to $top\_k = 10$. SARACODER achieves comprehensive superiority in Python tasks (e.g., *9.4% EM improvement*) while maintaining advantages in Java tasks despite a marginal $0.1$ *decrease in ES*. Crucially, under resource-constrained $top\_k = 1$ conditions: all Python metrics outperform the baseline; three Java metrics (EM/ES/ID_EM) show improvements; and Java ID_F1 initially trails (35.22 vs. 35.27) but ultimately surpasses the baseline at saturation (35.84 vs. 35.77). Our method achieves performance breakthroughs at lower computational cost (stable at $top\_k \approx 4$) by reducing redundant and homogeneous cases (Figure 4).

---

[2]For relevant explanations, please refer to the Appendix A.6.2

Table 2: Performance comparison on the CrossCodeEval dataset. Numbers are shown in percentage (%). The top results are **bolded**, and the second best are underlined.

| Language | Methods | Codegen2-7b | | | | Codegen25-7b | | | | deepseek-coder-6.7b-instruct | | | | CodeLlama-7b-Instruct | | | |
|---|---|---|---|---|---|---|---|---|---|---|---|---|---|---|---|---|---|
| | | Code Match | | Identifier Match | | Code Match | | Identifier Match | | Code Match | | Identifier Match | | Code Match | | Identifier Match | |
| | | EM | ES | EM | F1 | EM | ES | EM | F1 | EM | ES | EM | F1 | EM | ES | EM | F1 |
| Python | No Rag | 0.00 | 13.38 | 0.00 | 2.24 | 0.00 | 13.26 | 0.00 | 2.10 | 0.00 | 4.51 | 0.00 | 0.57 | 0.00 | 13.27 | 0.00 | 2.22 |
| | Shifted Rag | 4.84 | 46.67 | 11.48 | 42.72 | 7.40 | 48.88 | 14.09 | 44.62 | 8.19 | 50.18 | 14.77 | 46.69 | 6.91 | 49.12 | 13.60 | 45.12 |
| | Vanilla Rag | 9.48 | 50.97 | 17.15 | 47.81 | 12.39 | 53.92 | 23.61 | 53.14 | 13.00 | 54.00 | 26.63 | 55.77 | 11.45 | 52.93 | 19.23 | 50.49 |
| | Repocoder | 12.47 | 54.08 | 21.57 | 51.89 | 16.62 | 56.80 | 25.73 | **57.85** | 17.11 | 58.11 | 26.71 | 56.46 | 15.14 | 56.28 | 24.56 | 54.22 |
| | Graphcoder | 10.88 | 52.36 | 19.68 | 49.73 | 14.54 | 55.29 | 23.38 | 52.61 | 15.53 | 57.05 | 24.29 | 55.01 | 13.30 | 55.41 | 22.63 | 52.91 |
| | SARACODER | **15.04** | **56.03** | **24.44** | **54.68** | **18.36** | **58.30** | **27.28** | 56.22 | **19.72** | **59.93** | **28.52** | **58.26** | **17.91** | **58.37** | **27.77** | **56.82** |
| Java | No Rag | 1.03 | 21.79 | 0.64 | 16.86 | 1.50 | 21.77 | 24.78 | 24.78 | 6.40 | 35.79 | 10.42 | 32.90 | 0.93 | 20.83 | 1.96 | 16.17 |
| | Shifted Rag | 6.08 | 46.09 | 12.11 | 43.76 | 5.89 | 38.00 | 10.23 | 36.44 | 5.84 | **36.19** | 11.64 | 35.23 | 6.73 | 43.75 | 12.71 | 41.68 |
| | Vanilla Rag | 9.30 | **47.42** | 15.71 | **45.69** | 10.38 | 40.76 | 15.29 | 39.91 | 8.88 | 33.59 | 15.01 | 33.51 | 10.93 | 45.01 | 17.81 | 44.08 |
| | Repocoder | 10.71 | 41.83 | 16.18 | 41.51 | **12.16** | **42.38** | **17.63** | **41.69** | 9.58 | 34.25 | 15.76 | 34.13 | **13.23** | **46.01** | **19.87** | **45.14** |
| | Graphcoder | 8.13 | 45.18 | 14.35 | 43.32 | 8.42 | 36.77 | 12.85 | 35.84 | 7.39 | 32.34 | 12.76 | 32.05 | 8.51 | 40.57 | 14.96 | 39.57 |
| | SARACODER | **11.73** | 46.69 | **18.37** | 45.47 | 11.40 | 39.33 | 16.22 | 38.97 | **11.92** | 34.55 | **17.72** | 34.99 | 12.95 | 42.71 | 19.54 | 42.33 |

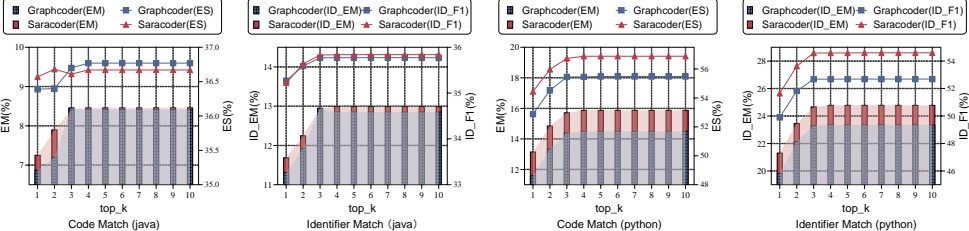

Figure 4: Impact of top_k on CrossCodeEval. (The two on the left are Java tasks, and the two on the right are Python tasks.)

### 4.2.3 FOR RQ3: SYNERGISTIC INTEGRATION OF SARACODER ACHIEVES ENHANCED COMPLETION.

We examine two prominent methods that demonstrate exceptional performance in Cross-File scenarios. **(1) Repocoder** Zhang et al. (2023), distinct from the original, assumes that if code snippets are similar, their subsequent content is also likely relevant. In the next search round, it specifically gets the code following those similar snippets (hereafter referred to as Repocoder). **(2) Draco** Cheng et al. (2024), analyzes code to create entity dependency graphs, allowing detailed background knowledge retrieval. It then uses this information to create structured prompts. Currently, Draco only works with Python. As shown in Table 3, adding our method significantly boosts all four Python metrics (by 3.42 to 4.52) compared to using Repocoder or Draco alone. For Java, our method improves EM by 0.45 and ID_EM by 0.33 over Repocoder, showing SARACODER exhibits significant synergistic gain property with existing cross-file methods. [3]

### 4.2.4 FOR RQ4: ENHANCED IN-FILE ACCURACY AND RESOURCE EFFICIENCY.

On the RepoEval-Updated dataset (Table 4), SARACODER shows superior semantic and identifier accuracy (surpassing the top-performing Graphcoder: +0.547 EM, +0.737 ES, +0.125 ID_EM, and +0.667 F1) for both Python and Java code completion. The cost analysis (Appendix A.6.1) further indicates SARACODER generally performs better and exhibits higher stability across most Python metrics (excluding EM) and all Java metrics. This makes it particularly effective for resource-constrained environments, especially at lower top_k values. However, SARACODER's gains over Graphcoder in code and identifier matching are smaller here than on CrossCodeEval. This is primarily because RepoEval-Updated projects contain a higher prevalence of similar code snippets, resulting in reduced code diversity within the repository. Overall, the conclusions align with those from the CrossCodeEval dataset.

### 4.3 ABLATION STUDY

To understand the importance of each part of SARACODER, we conduct ablation tests on the Cross-CodeEval dataset (Figure 5). "-EAID" indicates disabling **External-Aware Identifier Disambiguator**, resulting in the loss of external dependency integration capabilities; "-HF_OP" denotes remov-

---

[3]You can find the causes of synergistic gains in Appendix A.6.3.

Table 3: Performance benefits of SARACODER when integrated orthogonally with other cross-file approaches (%). The CrossCodeEval dataset is used in this part.

| Language | Methods | Codegen2-7b | | | | Codegen25-7b | | | | CodeLlama | | | |
|---|---|---|---|---|---|---|---|---|---|---|---|---|---|
| | | Code Match | | Identifier Match | | Code Match | | Identifier Match | | Code Match | | Identifier Match | |
| | | EM | ES | EM | F1 | EM | ES | EM | F1 | EM | ES | EM | F1 |
| Python | No Rag | 5.44 | 57.85 | 11.71 | 42.22 | 7.77 | 60.52 | 14.45 | 45.40 | 9.49 | 61.97 | 16.44 | 47.36 |
| | Shift Rag | 4.87 | 58.36 | 11.64 | 42.91 | 7.44 | 60.20 | 14.17 | 44.78 | 6.95 | 60.35 | 13.75 | 45.36 |
| | Vanilla Rag | 9.52 | 61.87 | 17.42 | 48.01 | 12.43 | 63.81 | 20.74 | 51.00 | 11.48 | 63.66 | 19.42 | 50.72 |
| | SARACODER | 12.16 | 54.16 | 18.37 | 45.47 | 11.50 | 44.90 | 16.22 | 38.95 | 13.32 | 48.04 | 19.54 | 42.34 |
| | Repocoder | 12.50 | 64.48 | 21.87 | 52.09 | 16.66 | 66.67 | 25.99 | 55.06 | 15.19 | 66.24 | 24.74 | 54.39 |
| | Repocoder + SARACODER | 15.94 +3.44 | 66.43 +1.95 | 25.73 +3.86 | 54.80 +2.71 | 19.49 +2.83 | 68.53 +1.86 | 28.90 +2.91 | 57.49 +2.43 | 19.19 +4.00 | 68.45 +2.21 | 29.05 +4.31 | 57.47 +3.08 |
| | Draco | 20.06 | 66.33 | 29.13 | 56.53 | 22.93 | 68.70 | 32.45 | 59.34 | 23.50 | 68.56 | 32.57 | 59.49 |
| | Draco + SARACODER | 24.06 +4.00 | 69.40 +3.07 | 34.00 +4.87 | 61.12 +4.59 | 27.05 +4.12 | 71.86 +3.16 | 36.80 +4.35 | 63.48 +4.14 | 27.20 +3.7 | 72.01 +3.45 | 37.29 +1.72 | 64.35 +4.85 |
| Java | No Rag | 0.00 | 25.92 | 0.05 | 17.48 | 0.00 | 25.46 | 0.05 | 17.61 | 0.00 | 25.17 | 0.00 | 17.23 |
| | Shift Rag | 6.45 | 54.84 | 12.11 | 43.75 | 6.08 | 44.73 | 10.27 | 36.46 | 7.11 | 50.96 | 12.72 | 41.68 |
| | Vanilla Rag | 9.68 | 55.71 | 15.71 | 45.71 | 10.47 | 47.09 | 15.29 | 39.93 | 11.31 | 51.48 | 17.81 | 44.09 |
| | SARACODER | 12.16 | 54.16 | 18.37 | 45.47 | 11.50 | 44.90 | 16.22 | 38.95 | 13.32 | 48.04 | 19.54 | 42.34 |
| | Repocoder | 11.22 | 56.89 | 17.72 | 47.41 | 10.85 | 47.93 | 16.18 | 41.50 | 13.60 | 52.17 | 19.87 | 45.14 |
| | Repocoder + SARACODER | 11.50 +0.28 | 56.09 -0.80 | 17.72 0.00 | 46.96 -0.45 | 11.27 +0.42 | 46.53 -1.40 | 16.41 +0.23 | 40.47 -1.03 | 14.26 +0.66 | 50.79 -1.38 | 20.62 +0.75 | 44.38 -0.76 |

Table 4: Performance comparison on the RepoEval-Updated dataset. Numbers are shown in percentage (%). The top results are **bolded**, and the second best are underlined.

| Language | Methods | Codegen2-7b | | | | Codegen25-7b | | | | deepseek-coder-6.7b-instruct | | | | CodeLlama-7b-Instruct | | | |
|---|---|---|---|---|---|---|---|---|---|---|---|---|---|---|---|---|---|
| | | Code Match | | Identifier Match | | Code Match | | Identifier Match | | Code Match | | Identifier Match | | Code Match | | Identifier Match | |
| | | EM | ES | EM | F1 | EM | ES | EM | F1 | EM | ES | EM | F1 | EM | ES | EM | F1 |
| Python | No Rag | 17.40 | 32.54 | 23.75 | 30.21 | 19.55 | 34.48 | 25.75 | 32.16 | 11.50 | 30.33 | 15.30 | 22.39 | 17.35 | 33.05 | 23.55 | 30.53 |
| | Shifted Rag | 32.70 | 59.22 | 40.10 | 55.66 | 36.45 | 61.96 | 43.20 | 58.31 | 20.90 | 42.95 | 26.50 | 38.88 | 33.90 | 60.28 | 41.50 | 56.39 |
| | Vanilla Rag | 38.70 | 63.58 | 46.45 | 60.43 | 42.25 | 66.26 | 48.75 | 62.79 | 22.20 | 41.48 | 27.85 | 37.58 | 40.30 | 65.03 | 47.55 | 61.06 |
| | Repocoder | 37.60 | 61.98 | 45.10 | 58.47 | 40.55 | 64.48 | 46.85 | 60.71 | 21.35 | 40.18 | 26.65 | 35.93 | 39.60 | 63.71 | 47.05 | 59.78 |
| | Graphcoder | 42.40 | 65.73 | 49.45 | 62.07 | **44.65** | 67.59 | **51.00** | 63.82 | **28.50** | 44.63 | 33.35 | 42.63 | 43.90 | 67.26 | 51.15 | 63.51 |
| | SARACODER | **42.60** | **65.92** | **50.15** | **62.61** | 44.50 | **67.79** | **51.10** | **63.84** | 28.25 | **46.91** | **33.45** | **42.95** | **45.00** | **68.27** | **52.00** | **63.97** |
| Java | No Rag | 6.55 | 16.84 | 9.15 | 8.84 | 5.35 | 16.21 | 9.05 | 8.65 | 6.40 | 20.61 | 7.75 | 8.73 | 6.85 | 17.11 | 9.45 | 9.06 |
| | Shifted Rag | 30.87 | 62.52 | 43.94 | 61.01 | 26.63 | 58.46 | 37.75 | 56.57 | 28.00 | 55.12 | 36.81 | 53.39 | 35.38 | 64.63 | 45.56 | 62.98 |
| | Vanilla Rag | 33.50 | 63.82 | 45.44 | 62.08 | 32.00 | 61.52 | 41.56 | 59.48 | 21.13 | 46.51 | 31.19 | 45.15 | 38.56 | 66.46 | 48.06 | 64.90 |
| | Repocoder | 30.13 | 60.01 | 42.31 | 57.10 | 28.75 | 57.73 | 37.88 | 54.46 | 22.19 | 46.93 | 32.06 | 44.26 | 35.38 | 62.33 | 44.44 | 59.63 |
| | Graphcoder | 37.75 | 66.19 | 50.68 | 64.77 | 36.63 | 64.74 | 46.13 | **62.62** | 28.81 | 55.75 | 40.06 | 53.61 | **43.00** | **69.68** | **52.69** | **67.85** |
| | SARACODER | **37.93** | **67.06** | **50.93** | **65.48** | **36.75** | **65.54** | **46.44** | 62.61 | **29.75** | **56.38** | **40.75** | **54.18** | 42.88 | 69.37 | 52.00 | 67.37 |

ing **Hierarchical Feature Optimization**, canceling the similar fragment screening mechanism; "-CCG" indicates disabling **the code context graph**, so it lost the understanding of code structure. The ablation experiments demonstrate that the complete SARACODER achieves optimal performance, with all components positively contributing to repository-level completion. Notably, even without EAID, SARACODER still outperforms Shift RAG and Vanilla RAG, and even surpasses Repocoder in Python tasks[4], proving that HF_OP screening substantially enhances case quality.

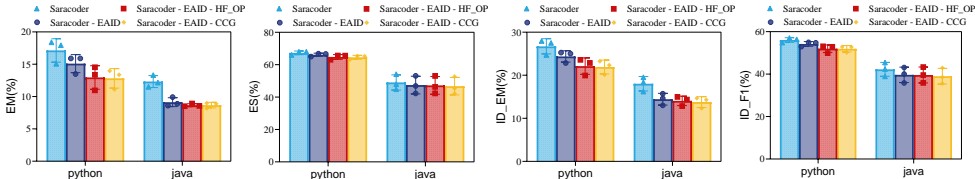

Figure 5: Ablation study. (Each three-data-point group represents CodeGen2-7B, CodeGen2.5-7B, and CodeLlama-7B-Instruct models. Bar lengths show their average performance, with I-shaped error bars indicating standard deviation)

## 5 CONCLUSION AND OUTLOOK

In this paper, we present SARACODER, a resource-optimized repository-level code completion method. It solves the problems of superficial similarity dispersion, retrieval redundancy and rigidity, and external symbol ambiguity by combining semantic topology with disambiguation. SARACODER uniquely addresses superficial similarity dispersion, retrieval redundancy and rigidity, and external symbol ambiguity, reducing unnecessary context window length consumption, and providing more diverse and higher-quality completion reference information content under resource-constrained conditions. This method improves code completion quality and can positively complement other cross-file methods, providing synergistic improvements when used in combination. However, while both Java and Python are prominent and widely used languages, the generalizability of this method to other programming languages has not yet been achieved. Future work will pursue two key directions: expanding language coverage and exploring cross-language code completion.

---

[4]Detailed data can be found in Table 8 in appendix.

## REPRODUCIBILITY STATEMENT

We have submitted the relevant code in the supplementary materials. The names of the experimental benchmarks, the prompt templates used, and the model's hyperparameter settings can all be found in Section 3.4 and A.5 . The Appendix A.5.1 and A.5.2 provides a detailed description of the experimental setup for the mechanism experiments.

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

# A APPENDIX

## A.1 CONTEXT GRAPH CONSTRUCTION

Code parsing transforms source code into an intermediate representation that is easier to analyze and process, and is a fundamental step to understand the semantics and structure of a program. Abstract

Syntax Tree (AST) is one of the most commonly used and effective intermediate representations in code parsing. It can map source code to a tree topology structure, and accurately represent the syntax features and context relationships of code elements. By traversing and manipulating the abstract syntax tree, the relationship graph in the code can be constructed efficiently. Tree-sitter is a CFG-based parser generator that can support a variety of programming languages including Python, Java, C++. The core advantage of Tree-sitter is its efficient parsing performance and wide support for multiple languages, which makes the system have the ability to parse the code of multiple programming languages uniformly, and provides the possibility of multi-language code analysis. In code, the following relationships play a key role in semantic analysis, refactoring, debugging, and maintenance. We use tree-sitter to model the following relationships. See Table 5 for details

Table 5: Semantic Relationships in Code Analysis

| Relationship | Definition | Syntax Examples | | Type-Sensitive Characteristics |
|---|---|---|---|---|
| | | Python | Java | |
| Assignment | Variable obtains type identity through assignment | `count: int = 10` | `String s = new String()` | Type inference and propagation |
| Contextual Binding | Creates temporary type bindings in specific syntactic structures | `with open(file) as f:` | `try (BufferedReader br = ...)` | Context-dependent type lifecycle management |
| Reference | Access to existing variables or properties | `obj.calculate()` | `this.value` | Late-bound type resolution |
| Type Declaration | Explicit annotation of variable/return types | `def func() -> list[str]:` | `List<Integer> list = new ArrayList<>()` | Basis for static type checking |
| Parameter Constraint | Type constraints on function parameters | `def sort(items: Sequence[T])` | `void sort(List<? extends Comparable> l)` | Input type validation |
| Return Constraint | Type constraints on function return values | `@return_type(float)` | `public int getValue() { ... }` | Output type consistency guarantee |
| Inheritance | Subclasses automatically acquire parent class members | `class Child(Parent):` | `class Child extends Parent { ... }` | Type hierarchy inheritance |
| Implementation | Class fulfillment of interface contracts | `class MyList(ABC):` | `class ArrayList implements List { ... }` | Foundation for polymorphic behavior |
| Override | Subclass overriding of parent class methods | `def method(self): ...` | `@Override void method() { ... }` | Dynamic method dispatch |
| Import Dependency | Cross-module import dependencies | `import pandas as pd` | `import java.util.List;` | Type visibility control |
| Invocation | Execution dependencies between methods/functions | `math.sqrt(x)` | `Collections.sort(list)` | Type compatibility verification |

## A.2 METHOD SUPPLEMENT

## A.3 IMPLEMENTATION OF CONTEXT GRAPH SLICING

We begin by initializing three empty sets: $V_{cf}$ for control flow, $V_{dd}$ for data dependencies, and $V_{cd}$ for control dependencies, along with an empty queue Q. The process starts by adding $v_{target}$ to Q. We then enter a loop, continuing as long as Q is not empty. In each iteration, a vertex v is dequeued. We apply two critical checks: first, a **hop count check** stops processing if v is more than h hops from $v_{target}$; second, a **size check** halts if the combined size of $V_{cf} \cup V_{dd} \cup V_{cd}$ reaches k statements. If v passes these checks, we update the sets: v goes into $V_{cf}$, its data dependency predecessors go into $V_{dd}$, and its control dependency predecessors go into $V_{cd}$. Following this, all unvisited control flow predecessors of v are enqueued. The loop concludes when Q is empty or a size/hop limit is hit. Finally, using the union $V_{cf} \cup V_{dd} \cup V_{cd}$ as the vertex set, we generate the **induced subgraph** $G_h(v_{target})$, which represents our final **context graph slice**.

## A.4 RETHINKING ON THE RETRIEVAL RANGE

Based on preliminary research, we observed that several retrieval-augmented methods for finding similar code snippets employ a zero-filtering strategy for subsequent code within the same file. Specifically, this strategy assigns a similarity score of zero—for both textual and graph-structural similarity—to any code segment located after the current line requiring completion within the same file, relative to the context of the current completion point. This approach stems from the assumption that code segments appearing after the completion point in the same file hold no semantic relevance to the current completion task. The rationale behind this assumption likely lies in a developer mind-set: "During code completion, neither the code being completed nor the subsequent code exists yet; therefore, later code offers no reference value."

We contend that this perspective may not universally hold. Firstly, in practical development, due to modular programming logic, most programmers' cognitive context is not continuous. Development does not strictly follow a top-down sequence based on physical line numbers; rather, it often involves non-linear thought processes. Consequently, there is no inherent correspondence between physical line numbers and the semantic boundaries of code segments. Secondly, within the same file, different functions may share similar implementations. For instance, Different classes might exhibit identical initialization logic; operations on distinct variables may follow similar patterns. Therefore, code segments following the completion point are not entirely devoid of reference value. In fact, due to the typically homogeneous nature of tasks handled within a single file, these subsequent segments might

be more contextually relevant to the actual scenario requiring completion compared to snippets retrieved from entirely different files.

### A.4.1 HE_OP'S PREVIOUS PREPARATION

After the database is established, when we receive a code completion request, we first conduct a global search for similar code snippet in the context of the location to be completed, and obtain a larger candidate pool by relaxing the location constraints. Use Jaccard similarity to calculate the text similarity with the query code, and select the $top\_k \times p$ samples with the highest similarity as the candidate set to participate in the subsequent process. Among them, Jaccard similarity is a measurement method used to measure the similarity between two sets. It measures the similarity between the two sets by calculating the ratio of their intersection to their union.

### A.4.2 THE ALGORITHM OF SEMANTIC ALIGNMENT DISTILLATION

For details, please refer to Algorithm 1.

---

**Algorithm 1** Code Similarity Analysis

---

**Require:** Query $Q$, Candidates $C$, Max length $L = 512$
**Ensure:** Filtered results $R$
  **Preprocess:**
  Tokenize and pad $Q$ and $C$ to length $L$
  **Encode:**
  **for** $s \in \{Q\} \cup C$ **do**
    Extract features $v_s \in \mathbb{R}^{768}$
    Normalize $v_s$
  **end for**
  **Filter:**
  Compute similarities $S = \{\cos(v_Q, v_c) | c \in C\}$
  Set threshold $\tau = \text{quantile}(S, 0.75)$
  $R \leftarrow \{c | \cos(v_Q, v_c) \geq \tau\}$
  **Cache:**
  **if** $s \in \text{Cache}$ **then**
    Retrieve cached $v_s$
  **else**
    Compute and store $v_s$
  **end if**

---

### A.4.3 D-SED CALCULATION

For details, please refer to Algorithm 2. The following is the symbol explanation.

- $X_{\mathcal{A}}$: Set of aligned vertices (mapped to target graph)
- $X_h^l(\hat{y}) \setminus X_{\mathcal{A}}$: Set of unaligned vertices (to be inserted/deleted)
- $E_{\mathcal{A}}$: Set of aligned edges
- $E_h^l(\hat{y}) \setminus E_{\mathcal{A}}$: Set of unaligned edges
- $h(v, \tilde{y})$: Distance from vertex $v$ to reference point $\tilde{y}$
- $c(v, \mathcal{A}(v))$: Vertex substitution cost (for aligned vertices)
- $c(v)$: Vertex insertion/deletion cost (for unaligned vertices)
- $c(e, \mathcal{A}(e))$: Edge substitution cost (for aligned edges)
- $c(e)$: Edge insertion/deletion cost (for unaligned edges)

**Key Features:**

- **Distance-based decay**: $\gamma^{h(v,\tilde{y})}$ weights edit costs based on proximity to reference

---

**Algorithm 2** Decay Subgraph Edit Distance (D-SED)

---

**Require:** Graphs $G_h^l(\hat{y})$ and $G_h^l(x)$
    Decay factor $\gamma \in (0, 1)$
**Ensure:** SED between $G_h^l(\hat{y})$ and $G_h^l(x)$
1: SED $\leftarrow 0$
2: **for** each vertex $v \in X_{\mathcal{A}}$ **do**
3:     SED $\leftarrow$ SED $+ \gamma^{h(v,\tilde{y})} \cdot c(v, \mathcal{A}(v))$
4: **end for**
5: **for** each vertex $v \in X_h^l(\hat{y}) \setminus X_{\mathcal{A}}$ **do**
6:     SED $\leftarrow$ SED $+ \gamma^{h(v,\tilde{y})} \cdot c(v)$
7: **end for**
8: **for** each edge $e = (v, t, u) \in E_{\mathcal{A}}$ **do**
9:     SED $\leftarrow$ SED $+ \gamma^{h(v,\tilde{y})} \cdot c(e, \mathcal{A}(e))$
10: **end for**
11: **for** each edge $e = (v, t, u) \in E_h^l(\hat{y}) \setminus E_{\mathcal{A}}$ **do**
12:     SED $\leftarrow$ SED $+ \gamma^{h(v,\tilde{y})} \cdot c(e)$
13: **end for**
14: **return** SED

---

- **Four cost categories**: Separates vertex/edge and aligned/unaligned cases
- **Reference point**: $\tilde{y}$ serves as the anchor for distance calculations
- **Asymmetric treatment**: Focuses on edits in $G_h^l(\hat{y})$ relative to $G_h^l(x)$

## A.5 DETAILS OF EXPERIMENT SETUP

### A.5.1 PARAMETER SETTING

In this study, we mainly adopt the Greedy Decoding strategy for text generation. Its core configuration is: disable sampling (`do_sample=False`) and set the temperature value (`temperature`) to 0. This combination ensures that the model necessarily selects the token with the highest logical probability at each step, thereby eliminating randomness in the generation process and facilitating a strict and reproducible evaluation of the model's performance. To simplify and observe the performance of various methods within the limited length of the context window, the maximum length limit for text is 2048 new tokens (`max_num_tokens`).

### A.5.2 RESOURCE CONTROL AND ALLOCATION IN THE EXPERIMENT

In practice, the `max_num_tokens` parameter controls the total length of the context window. The system sets a maximum length limit for the "unfinished code context," which is no more than half of the total window. The remaining context window is dynamically used to accommodate other information, including retrieved similar code snippets. Since the output of the External Symbols Enhancement module is highly compressed, we primarily use the *top_k* parameter to control the number of similar code snippets that can be included in the prompt. This effectively constrains the use of this portion of the resources. The value of *top_k* directly determines the number of similar examples that can be introduced; a smaller value allocates less context space to similar code snippets. Through this mechanism, we can achieve flexible and precise control over the different components of the information within a limited total resource budget.

### A.5.3 DETAILS OF DATASET

The detailed information of dataset is as follows in the table 6 and table 7.

### A.5.4 CALCULATION OF EVALUATION INDICATORS

(1) Code Exact Match(EM): The code exact match measures the proportion of the generated code completions that are exactly the same as the ground truth.

Table 6: The dataset of CrossCodeEval

|  | **Python** | **Java** |
|---|---|---|
| Total number of repositories | 471 | 239 |
| Total number of documents | 14348 | 5868 |
| Total number of task cases | 2665 | 2139 |

Table 7: The dataset of RepoEval-Updated

| **Language** | **Project Name** | **Creation time** | **The number of files** | **Total project file size (MB)** |
|---|---|---|---|---|
| Python | devchat | 2023-04-17 | 40 | 0.5 |
|  | nemo_aligner | 2023-09-01 | 54 | 1.6 |
|  | awslabs_fortuna | 2022-11-17 | 168 | 1.9 |
|  | task_weaver | 2023-09-11 | 113 | 3.0 |
|  | huggingface_diffusers | 2022-05-30 | 305 | 6.2 |
|  | opendilab_ACE | 2022-11-23 | 425 | 6.8 |
|  | metagpt | 2023-06-30 | 374 | 17.9 |
|  | apple | 2023-02-25 | 265 | 23.8 |
|  | QingruZhang | 2023-05-31 | 1357 | 32.6 |
|  | nerfstudio-project_nerfstudio | 2022-05-31 | 157 | 54.5 |
| Java | FloatingPoint-MC_MIN | 2023-07-10 | 2628 | 269.5 |
|  | itlemon_chatgpt4j | 2023-04-04 | 67 | 0.4 |
|  | mybatis-flex_mybatis-flex | 2023-02-27 | 487 | 8.8 |
|  | Guiqu1aixi_rocketmq | 2023-04-25 | 988 | 10.6 |
|  | SimonHalvdansson_Harmonic-HN | 2023-05-23 | 51 | 16.8 |
|  | Open-DBT_open-dbt | 2023-02-27 | 366 | 20.0 |
|  | QuasiStellar_custom-pixel-dungeon | 2023-05-08 | 1093 | 51.3 |
|  | gentics_cms-oss | 2023-05-08 | 2580 | 130.5 |

(2) Identifier Exact Match(ID_EM): The identifier exact match measures the proportion of the generated code completions that are exactly the same as the ground truth.

(3) Identifier F1 Score: The Identifier F1 score measures the degree of match between the identifiers (variable names, function names, etc.) in the generated code and the actual identifiers. It combines Precision (correctness) and Recall (completeness).

$$F1 = 2 \times \frac{\text{precision} \times \text{recall}}{\text{precision} + \text{recall}} \tag{3}$$

(4) Edit Distance Similarity (ES) : Edit Distance similarity is calculated based on the edit distance and measures the degree of similarity between the generated code string and the real code string.

$$ES = 1 - \frac{ED(S_1, S_2)}{\max(\text{len}(S_1), \text{len}(S_2))} \tag{4}$$

Among them, $\text{len}(S_1)$ and $\text{len}(S_2)$ are the lengths of $S_1$ and $S_2$ respectively. $ED(S_1, S_2)$ is the edit Distance between $S_1$ and $S_2$, also known as the Levenshtein Distance(Algorithm 3), which is usually calculated through Dynamic Programming.

### A.5.5 DETAILS OF BASELINES

- **No RAG:** As a basic control experiment, this method only relies on the pre-trained knowledge base of large language models (LLMS), and directly inputs the current code context into the model for autoregressive generation. The characteristic of this method lies in completely ignoring the context information of the code base, and it can be used to evaluate the native reasoning ability of LLMS in zero-shot scenarios.

- **Shifted RAG:** The core of this method is the sliding window offset mechanism. This mechanism dynamically adjusts window positions during retrieval, prioritizing code segments likely to contain target call chains. Through temporal probability prediction, it enhances temporal relevance between retrieval results and completion targets. The approach demonstrates distinct advantages in scenarios like API invocation sequences and control flow continuation.

- **Vanilla Rag:** Given the context, retrieve a set of similar code snippets from the repository through a fixed-size sliding window and call the LLM to obtain the predicted next statement.

---

**Algorithm 3** Levenshtein Distance Calculation

---

**Require:** Two strings: $str1$ (length $m$), $str2$ (length $n$)
**Ensure:** Edit distance between $str1$ and $str2$
  Initialize $dp$ as 2D array of size $(m+1) \times (n+1)$
  **for** $i = 0$ **to** $m$ **do**
    $dp[i][0] \leftarrow i$ {Deletion operations}
  **end for**
  **for** $j = 0$ **to** $n$ **do**
    $dp[0][j] \leftarrow j$ {Insertion operations}
  **end for**
  **for** $i = 1$ **to** $m$ **do**
    **for** $j = 1$ **to** $n$ **do**
      **if** $str1[i-1] == str2[j-1]$ **then**
        $dp[i][j] \leftarrow dp[i-1][j-1]$ {Characters match}
      **else**

$$dp[i][j] \leftarrow 1 + \min \begin{cases} dp[i][j-1] & \text{(Insertion)} \\ dp[i-1][j] & \text{(Deletion)} \\ dp[i-1][j-1] & \text{(Substitution)} \end{cases}$$

      **end if**
    **end for**
  **end for**
  **return** $dp[m][n]$ {Final edit distance}

---

- **Repocoder**[5]**:** This is an iterative retrieval-augmented framework for repository-level code completion. It addresses the challenge of leveraging fragmented repository information by integrating similarity-based retrievers with pretrained code LLMs, enabling precise cross-file completion of unfinished code.

- **Graphcoder:** This is a structured retrieval-augmented code completion framework. Its core innovation lies in employing a graph-based retrieval-generation process, which utilizes Code Context Graphs (CCG) to accurately model code dependencies, replacing traditional sequence-based context representations.

## A.6 ADDITIONAL RESULTS

Table 8: Detailed data of the ablation experiment

| Language | Methods | Codegen2-7b | | | | Codegen25-7b | | | | CodeLlama | | | |
| | | Code Match | | Identifier Match | | Code Match | | Identifier Match | | Code Match | | Identifier Match | |
| | | EM | ES | EM | F1 | EM | ES | EM | F1 | EM | ES | EM | F1 |
|---|---|---|---|---|---|---|---|---|---|---|---|---|---|
| Python | No Rag | 5.44 | 57.85 | 11.71 | 42.22 | 7.77 | 60.52 | 14.45 | 45.40 | 9.49 | 61.97 | 16.44 | 47.36 |
| | Shift Rag | 4.87 | 58.36 | 11.64 | 42.91 | 7.44 | 60.20 | 14.17 | 44.78 | 6.95 | 60.35 | 13.75 | 45.36 |
| | Vanilla Rag | 9.52 | 61.87 | 17.42 | 48.01 | 12.43 | 63.81 | 20.74 | 51.00 | 11.48 | 63.66 | 19.42 | 50.72 |
| | SARACODER | 15.07 | 66.04 | 24.71 | 54.86 | 18.40 | 67.95 | 27.50 | 56.38 | 17.94 | 67.99 | 27.96 | 57.00 |
| | - EAID | 13.49 | 65.02 | 22.86 | 52.95 | 15.90 | 66.80 | 25.01 | 54.74 | 15.90 | 66.63 | 25.27 | 54.98 |
| | - HF_OP | 10.96 | 63.05 | 19.95 | 49.90 | 14.54 | 65.66 | 23.57 | 52.88 | 13.37 | 65.63 | 22.86 | 53.11 |
| | - CCG | 11.11 | 63.23 | 20.14 | 50.19 | 13.98 | 65.27 | 23.27 | 52.78 | 13.37 | 65.28 | 22.36 | 52.86 |
| Java | No Rag | 0.00 | 25.92 | 0.05 | 17.48 | 0.00 | 25.46 | 0.05 | 17.61 | 0.00 | 25.17 | 0.00 | 17.23 |
| | Shift Rag | 6.45 | 54.84 | 12.11 | 43.75 | 6.08 | 44.73 | 10.27 | 36.46 | 7.11 | 50.96 | 12.72 | 41.68 |
| | Vanilla Rag | 9.68 | 55.71 | 15.71 | 45.71 | 10.47 | 47.09 | 15.29 | 39.93 | 11.31 | 51.48 | 17.81 | 44.09 |
| | SARACODER | 12.16 | 54.16 | 18.37 | 45.47 | 11.50 | 44.90 | 16.22 | 38.95 | 13.32 | 48.04 | 19.54 | 42.34 |
| | - EAID | 8.88 | 53.12 | 14.63 | 43.12 | 8.60 | 42.29 | 13.00 | 35.84 | 9.91 | 46.89 | 15.71 | 40.03 |
| | - HF_OP | 8.56 | 53.08 | 14.35 | 43.33 | 8.51 | 42.29 | 12.86 | 35.84 | 8.93 | 46.36 | 14.96 | 39.57 |
| | - CCG | 8.88 | 52.67 | 14.31 | 42.92 | 8.18 | 42.10 | 12.34 | 35.57 | 8.98 | 45.44 | 14.68 | 38.51 |

---

[5]We attempt to run the code publicly released by Zhang et al., but fail to execute them with the provided instructions. For the specific implementation here, please refer to Ding et al.

### A.6.1 COST ANALYSIS IN IN-FILE SCENARIOS

In this Token cost analysis experiment on the efficiency of code completion inference, we used the RepoEval_Updated dataset, the Deepseeking -coder model for code inference, and Graphcoder as the control group for the experiment. The main comparison is to show the changes of each accuracy index as the number of similar cases retrieved by the code in the prompt word (top-k) increases. As can be seen from Figure 6, whether it is a Graphcoder or SARACODER, with the increase of top_k, they basically show a trend of first rising, then falling, and finally gradually stabilizing. This indicates that as top-k increases, the noise cases that may be introduced may lead to a decrease in accuracy. SARACODER demonstrates advantages in the vast majority of scenarios: in the python code completion task, the stable values of the three metrics except EM are higher than those of the original retrieval method, among which the ES value increases by 0.31, the ID_EM value increases by 0.1, and the ID_F1 value increases by 0.32. In the java code completion task, the stable values of the four indicators have all seen relatively significant improvements, increasing by 0.937, 0.641, 0.687, and 0.568 respectively. In addition, it can be seen from the figure that when the top_k is between 2 and 4, the performance of the improved retrieval method is significantly better than that of the original retrieval method. This indicates that our method is more suitable for scenarios with scarce computing resources.

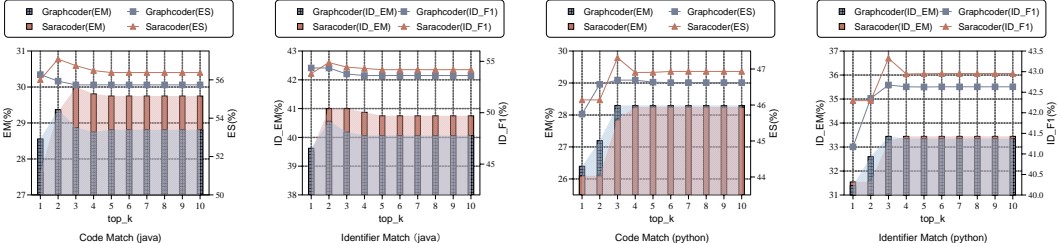

Figure 6: Impact of top_k on RepoEval-Updated. (The two on the left are Java tasks, and the two on the right are Python tasks.

### A.6.2 QUANTITATIVE ANALYSIS OF TOP_K AND TOKEN.

In our experiments, we focused on minimizing the influence of external factors. To do this, we used the smallest runnable (executable and producing valid output) module, which included both unfinished code context and HE_OP (Hierarchical Feature Optimization). The unfinished code context provided the input content, while HE_OP offered code completion reference cases. Importantly, HE_OP is directly influenced by the top_k parameter. We conducted our tests using two key top_k values(4 and 10) and set max_token_num to 2048. As shown in the table 9, clearly demonstrate that a top_k of 4 significantly reduces input token consumption across all three models compared to a top_k of 10. On average, each task saved approximately 22.38 input tokens when top_k was set to 4, confirming that a smaller top_k value leads to lower token consumption. Furthermore, we observed no significant drop in output token count when top_k was reduced. This, coupled with the results in Figure 6 showing no decline in accuracy, indicates that our method effectively reduces resource consumption while maintaining output quality.

Table 9: Quantitative analysis of top_k and token. The comparison of the average input and output tokens on each task when the dataset is Repo_Updated and max_num_tokens is 2048.

| Method | Codegen2-7b | | deepseek-coder-6.7b-instruct | | CodeLlama-7b-Instruct | |
|---|---|---|---|---|---|---|
| | #In | #Out | #In | #Out | #In | #Out |
| context | 821.627 | 95.15 | 778.52 | 93.48 | 763.93 | 96.86 |
| context+HF_OP (top_k=10) | 1665.50 | 88.15 | 1621.92 | 90.86 | 1611.37 | 96.99 |
| context+HF_OP (top_k=4) | **1644.26** | 89.20 | **1596.83** | 88.57 | **1590.55** | 98.06 |

### A.6.3 CASE STUDY IN SYNERGISTIC GAIN PROPERTY

To better understand the causes of synergistic gains, we analyzed our experimental cases. SARA-CODER's External-Aware Identifier Disambiguator effectively resolves "type inference chain break-

age" in cross-file dependencies by injecting essential symbolic relationships. Still, it occasionally introduces irrelevant information, which can lead to misinterpretations. Repocoder, when used independently, offers prompts that closely align with common coding patterns. However, it faces challenges with the adaptability of external information, often clashing with local APIs or project constraints, thereby limiting its accuracy. Draco stands out for its deep semantic modeling, which generates detailed data and control flow graphs to pinpoint highly relevant cross-file context; nonetheless, it encounters difficulties when the code's intent is unclear. SARACODER significantly contributes by providing semantically aligned code examples. These examples offer crucial "intent hinting" and "structure references," compensating for Draco's limitations in ambiguous code scenarios. As a result, the "Draco + SARACODER" combination synergistically boosts performance: Draco delivers precise cross-file context, while SARACODER guides intent and structure. Moreover, the external disambiguation module within SARACODER clarifies identifiers, effectively alleviating Repocoder's issues with external information adaptability and conflicts, making the "Repocoder + SARACODER" combination a more effective choice than using Repocoder in isolation.

A.7    THE USE OF LARGE LANGUAGE MODELS (LLM)

In order to enhance the language quality and clarity of this academic paper, the author utilized AI-powered tools for text refinement during the writing process. The specific details are as follows:

**Purpose of Use:** The primary purposes for using AI tools were to:

- Check grammar and spelling for certain sentences.
- Optimize vocabulary choices for more precise and academic expression.
- Adjust sentence structures to improve logical coherence and readability between paragraphs.

**Method of Use:** The author input original paragraphs written by themselves into the AI tools and then manually judged, filtered, and revised the text based on the refinement suggestions provided. All adopted changes were carefully considered by the author to ensure they fully align with the original intent and academic rigor of the paper.

**Disclaimer of Responsibility:** All academic content in this paper, including core arguments, research data, result analysis, argumentation process, and final conclusions, was independently created and is the sole responsibility of the author. The AI tools were used purely as an auxiliary aid and did not generate any critical academic viewpoints, research data, or conclusions. The author assumes full responsibility for the final content of the paper.

**Tools Used:** The AI tools used in this process were: Gemini-2.5 Flash, deepseek-V3.1.

