# OpenReview forum: "SaraCoder: Orchestrating Semantic and Structural Cues for Resource-Optimized Repository-Level Code Completion"
_ICLR.cc/2026/Conference — ICLR 2026 Conference Withdrawn Submission_

### Official Review · Reviewer_GUk7 · 2025-10-17

**Soundness:** 2
**Presentation:** 3
**Contribution:** 2
**Rating:** 4
**Confidence:** 4

**Summary:**

The paper proposes SARACODER, a resource-optimized retrieval-augmented framework for repository-level code completion. It introduces a Hierarchical Feature Optimization module that filters and reranks retrieved code snippets by semantic alignment, redundancy pruning, structural similarity, and diversity, as well as an External-Aware Identifier Disambiguator to resolve cross-file symbol ambiguity. Experiments on CrossCodeEval and RepoEval show that SARACODER performs well and consistently outperforms baselines such as GraphCoder and RePoCoder in Python, improving EM and ES scores across two model families.

**Strengths:**

+ The proposed hierarchical optimization pipeline is well-motivated and systematically integrates multiple complementary criteria (semantic alignment, redundancy reduction, structural similarity, and diversity).

-The paper is clearly written, and the methodology is easy to follow

**Weaknesses:**

My main concerns are as follows:

1. **Limited conceptual novelty.**
   The paper is largely incremental. It stacks multiple retrieval and context-filtering modules (semantic filtering, reranking, enhancement, etc.) on top of an existing RAG framework. While the overall pipeline is systematic, similar ideas have already been explored in prior works. The paper lacks clear theoretical insight or conceptual novelty beyond combining known heuristics.

2. **Limited empirical significance and questionable scalability.**
   The reported performance gains appear modest, despite the additional complexity introduced by the multi-stage pipeline. More importantly, all experiments are restricted to relatively small models (≤7B). Given that current state-of-the-art commercial and open-source code models already demonstrate strong repository-level capabilities, it remains unclear whether the proposed context optimization is still necessary or effective for larger models. This question is crucial to determine the true impact of the work.

3. **Pipeline complexity without clear analysis.**
   Although Section 4.3 includes an ablation study, the hierarchical optimization framework involves four distinct components, yet their individual contributions, interactions, and potential redundancy are not well analyzed. The “Code Context Graph” mentioned in the experiments is also unclear, as it does not correspond to any well-defined module in the methodology. Parameters such as the *dynamic threshold* in Section 3.2.1 and the *λ* term in Section 3.2.4 lack explanation or rationale for their selection.

4. **Efficiency not convincingly demonstrated.**
   The authors describe the method as “resource-optimized,” but the design introduces several extra steps. The paper only discusses a top-k study without providing analysis of time overhead, throughput, or resource consumption. Without such evidence, it is difficult to justify the claimed efficiency.


5. **inconsistencies in reported results.**
   Tables 2, 3, and 8 all report results on *CrossCodeEval* with the same model, but the EM and ES scores for SaraCoder vary substantially across them. In Table 3, SaraCoder even performs worse than vanilla RAG, and the Python and Java results appear identical. It is unclear whether this is a typo, data mismatch, or missing explanation, and it should be clarified.

6. **Lack of deeper insight.**
   The evaluation focuses solely on EM and ES, which are not very strong or informative metrics for code generation. More insightful analysis is missing—for instance, when and why the proposed framework succeeds or fails across different retrieval scenarios.

**Questions:**

**Questions for Authors**

1. Since your framework does not involve any training or parameter tuning of the base model, could you provide additional results on larger or commercial code models (e.g., GPT-4, Claude, or CodeLlama-70B) to demonstrate whether the proposed optimization still brings improvements? This would help establish the generality and scalability of your method.

2. In Section 4.3, it is mentioned that a *Code Context Graph* is used, but it is unclear which module in your framework produces or consumes this graph. Could you clarify how this graph is constructed and how it fits into your overall pipeline?   Relatedly, in Section 3.2.3, you refer to a *query graph* for an unfinished piece of code. How exactly is this query graph obtained, given that the code is incomplete?

3. Could you explain the rationale behind the hyperparameter settings—specifically, the *dynamic threshold* in Section 3.2.1 and the *λ* term in Section 3.2.3? How were these values chosen, and how sensitive is your method to them?

---

### Official Review · Reviewer_hF3h · 2025-10-28

**Soundness:** 2
**Presentation:** 2
**Contribution:** 3
**Rating:** 2
**Confidence:** 4

**Summary:**

This paper proposes SaraCoder, a retrieval-augmented approach for repository-level code completion, which aims to improve context efficiency by filtering and reranking retrieved code snippets based on semantic and structural cues. SaraCoder combines deduplication, graph-based structural similarity, and identifier disambiguation to enhance diversity and relevance within limited context windows. It reports improved completion accuracy on cross-file and in-file benchmarks.

**Strengths:**

The paper focuses inefficient context utilization in retrieval-augmented generation, which is a timely and practical issue in repository-level code completion. The motivation is sound:  reducing redundancy and increasing information diversity within constrained context windows can improve both model performance and system efficiency.

**Weaknesses:**

1. Clarity and Organization: Key concepts such as program slicing, query graph, and candidate graph are introduced without sufficient detail in the main body, and there is no clear cross-reference to their definitions in the appendix. Conversely, details like the use of GraphCodeBERT or MD5 hashing are elaborated unnecessarily in the main body, distracting from the high-level contributions.

2. System Complexity and Focus: SaraCoder comprises numerous components, making it more suitable for software engineering. As a result, it demands a thorough, fine-grained ablation study.

3. Insufficient Efficiency Analysis: Despite claiming to be "resource-optimized," the paper fails to provide direct comparisons of token consumption or latency. Specifically, (1) How does the input token count of SaraCoder compare to baselines? (2) What is the overhead of running individual modules? Prompt construction time is not reported, yet this directly impacts real-time usability. (3) Results in Tables 2 and 3 are not consistent, and the values of top_k are not specified.

4. Inadequate Related Work Coverage: The authors overlook several relevant prior works that have explored similar ideas: The fusion of semantic and structural cues has been investigated [1,2,3], especially RepoFuse [1]. Context pruning and compression techniques [4,5] are highly relevant to the claimed contribution but are not discussed or compared.

[1] RepoFuse: Repository-Level Code Completion with Fused Dual Context

[2] GRACE: Graph-Guided Repository-Aware Code Completion through Hierarchical Code Fusion

[3] CodeRAG: Finding Relevant and Necessary Knowledge for Retrieval-Augmented Repository-Level Code Completion

[4] Hierarchical Context Pruning: Optimizing Real-World Code Completion with Repository-Level Pretrained Code LLMs

[5] LongCodeZip: Compress Long Context for Code Language Models

**Questions:**

See Weaknesses.

---

### Official Review · Reviewer_ReX2 · 2025-10-28

**Soundness:** 1
**Presentation:** 3
**Contribution:** 2
**Rating:** 2
**Confidence:** 2

**Summary:**

This paper proposed SaraCoder for repo-level code completion. This paper mainly aims to solve the problem of redundant and highly similar texts in retrieved contexts. To solve this problem, the authors propose Hierarchical Feature Optimization that applies semantic alignment distillation, redundancy-aware pruning, and diversity-aware reranking. Extensive experiments on CrossCodeEval and RepoEval-Updated demonstrate the effectiveness of SaraCoder.

**Strengths:**

1. The paper is well written and easy to follow.
2. The improvement of EM and ES on two datasets demonstrates the effectiveness of SaraCoder.

**Weaknesses:**

1. The motivation of this paper is doubtful. According to the authors, one of the problems in current RAG systems is that the retrieved codes are highly similar to each other. To solve this problem, the authors propose HF_OP, which includes redundancy elimination and diversity-aware reranking. And the authors give an illustrative example in Figure 1(b). However, I doubt the frequency of this phenomenon. Usually, if a project is well-designed, developers seldom reinvent the wheel. The authors should justify this point.
2. Following the first point, I find that there are no ablation studies that only eliminate HF_OP in Figure 5, which is an important ablation design. Otherwise, we can not see the effectiveness of solving the problem that the authors want to address in this paper.
3. The proposed SaraCoder incorporates many additional preprocessing steps, compared to baseline methods. Yet, there is no time cost comparison.
4. I carefully check the experimental results in Table 2 and Table 8. They are all evaluated on CrossCodeEval. However, I find misaligned experimental results. For example, The ES of Codegen2-7b on CrossCodeEval-Python without RAG is 13.38 in Table 2, and it is 57.85 in Table 8. This is not the only case. Please give explanations for this mismatch. This makes me doubt the solidity of all the experimental results in this paper.

**Questions:**

Please see my comments above.

---

### Official Review · Reviewer_PXQt · 2025-11-01

**Soundness:** 2
**Presentation:** 2
**Contribution:** 2
**Rating:** 2
**Confidence:** 4

**Summary:**

The paper presents SARACODER, a repository-level code completion framework combining multiple retrieval and filtering techniques: semantic alignment (GraphCodeBERT-based), structural similarity (graph edit distance), redundancy pruning (MD5 hashing), and external symbol disambiguation (EAID). It aims to optimize retrieval diversity and efficiency within a constrained context window. Experiments on CrossCodeEval and “RepoEval-Updated” show modest gains over existing RAG baselines such as GraphCoder and Repocoder.

**Strengths:**

- Comprehensive system design: The paper systematically covers multiple aspects of retrieval-based code completion — semantic similarity, structure, redundancy, and external symbol handling.

- Practical focus: Addresses real-world concerns such as resource constraints and redundant retrievals in repo-level code completion.

**Weaknesses:**

- Limited novelty: Each module (semantic filtering, code graph construction, redundancy reducing, reranking) has been well-studied in prior research work. The paper seems to combines known components without a new conceptual insight or methodological breakthrough.

- Weak significance: The improvements are incremental and largely driven by EAID, which itself is a straightforward dependency lookup. The other “hierarchical optimization” modules add marginal effect.

- Benchmark ambiguity: “RepoEval-Updated” is introduced but not clearly defined—what was updated, and why not just use RepoEval? This lack of transparency raises questions about generalization and reproducibility.

- Missing motivation and clarity: Section 3.2 lists modules mechanically, but there are no concrete motivating examples showing why each module matters or how it improves code quality. Without such context, the design feels heuristic.

- Shallow ablations: The ablation study disables large modules at once, offering little insight into the incremental value of sub-components.

**Questions:**

- What exactly was modified in RepoEval-Updated, and does it change the difficulty or distribution of the tasks?

- What are the individual contributions to the final performance of the proposed HF_OP modules?

---

### Note · Authors · 2026-01-05

I have read and agree with the venue's withdrawal policy on behalf of myself and my co-authors.